# The Structure and Mechanical Properties of the Surface Layer of Polypropylene Polymers with Talc Additions

**DOI:** 10.3390/ma13030698

**Published:** 2020-02-04

**Authors:** Michał Świetlicki, Dariusz Chocyk, Tomasz Klepka, Adam Prószyński, Anita Kwaśniewska, Jarosław Borc, Grzegorz Gładyszewski

**Affiliations:** 1Department of Applied Physics, Lublin University of Technology, ul. Nadbystrzycka 38, 20-618 Lublin, Poland; d.chocyk@pollub.pl (D.C.); a.proszynski@pollub.pl (A.P.); a.kwasniewska@pollub.pl (A.K.); j.borc@pollub.pl (J.B.); g.gladyszewski@pollub.pl (G.G.); 2Department of Polymer Processing, Lublin University of Technology, ul. Nadbystrzycka 36, 20-618 Lublin, Poland; t.klepka@pollub.pl

**Keywords:** polypropylene, hardness, elastic modulus, friction, talc, X-ray diffraction

## Abstract

In the presented work the influence of different 3MgO·4SiO_2_·H_2_O (talc) contents in polypropylene samples on the structure, hardness, elasticity, and friction of the surface layer was investigated. The talc content ranged from 0 to 25 wt.%, and all the samples were obtained in the same conditions by the injection molding process. The analysis of the microstructure was performed by X-ray diffraction. Changes in the hardness and elasticity were determined for three different depths (300, 800, and 4000 nm) using an ultra nano tester. For the purpose of the examination of the friction properties of the obtained compounds, a nano-scratch tester was applied. Increasing the talc content caused growth in the indentation modulus and hardness values. Simultaneously, an effect of decreasing hardness and elastic modulus with increasing indentation depth was observed. The smallest effect size was observed for 25 wt.% talc content, which might suggest that talc addition increased the homogeneity of the observed composites. Scratch tests showed increasing scratch resistance along with increasing talc content for both constant and progressive loads. The growth in talc concentration led to a decrease in the degree of the polypropylene (PP) crystallinity of the surface layer. The exfoliation process occurred in PP composites.

## 1. Introduction

Micromachine parts, such as gears, pulleys, and plain bearings are usually made of polyamide (PA). PA, as well as being rigid and melt resistant, belongs to a group of hygroscopic substances [1], which negatively affects the operation of devices made of this material in the presence of water. Therefore, it is reasonable to look for the components made of materials that are free of this defect and with similar or even better mechanical and sliding properties. In humid environments one of the potential substitutes for polyamide, in a wide range of applications, is polypropylene (PP) [2,3,4,5].

Polypropylene composite is a widely produced and used polymer applied, e.g., in the construction, automotive, packaging, cable-insulation, and household-goods industries [6]. Isotactic PP is notch sensitive and brittle under severe conditions of deformation, such as low or high temperatures, which leads to its reduced usage and limitations in production lines [6,7]. Accordingly, there is a strong need for improvements to PP’s mechanical and thermal properties, mainly by the incorporation of different fillers, especially since polypropylene is characterized by good processability and accepts different types of natural and synthetic fillers [8,9,10,11]. Mica, kaolin, calcium carbonate, and talc are the fillers used most often, with the aim of reducing production costs and improving the properties of the thermoplastics, such as rigidity, strength, hardness, flexural modulus, dimensional stability, crystallinity, and electrical and thermal conductivity [7,12].

The type of filler, its content and size, interfacial adhesion, and the bond strength between the PP matrix and the filler, as well as the surface characteristics of the composite, can greatly influence the filled system [13], changing its final characteristics. It is known, for example, that mineral fillers used to alter the properties of polypropylene can adversely affect the mechanical parameters of the composites. The presence of talc in thermoplastics significantly decreases impact strength but usually results in improved tensile and flexural properties [7,13,14,15,16,17].

The top layer of the material is responsible for the external characteristics of the product, which include increased hardness, scratch resistance, and improved wear resistance [18]. Machine parts made of PP can undergo cutting and fragmentation. The determination of particular types of damage occurring on the surface and the development of a method to identify a set of criteria to predict the type of damage features during a production process have implications in areas where polymers are used as structural or coating materials [19]. Studies that have been conducted to assess the influence of surface quality on the mechanical behavior of reinforced polymer materials have mainly been focused on the surface roughness [18,20] and modification of the crystalline structure of the polymer [21]; however, changes to the mechanical parameters of talc-filled polypropylene with an increasing indentation depth have not yet been described.

Well-aligned and reliable methods for surface analysis also applied in the characterization of polymers [22,23,24] are the scratch and indentation tests. To characterize the surface layer itself, mechanical tests need to be conducted at relatively small depths and with relatively low forces.

In the case of studies on the structure and mechanical properties of surface layers, a phenomenon related to the dependence of mechanical parameters of the surface layer on the size of indentations must also be considered. The indentation size effect (ISE) was revealed in nanoindentation experiments, which showed the dependence of the mechanical behavior of various materials, including polymers, on the indentation depth [24]. The above-mentioned relationship is associated with changes to deformation mechanisms along with the length scale [25,26]. In polymers and polymer composites, the problem of the non-uniformity of properties has a multifactorial origin. It has been suggested that it results from higher-order displacement-gradient models [27,28,29], surface effects [30], and friction [31]. It has also been suggested that it is dependent on the heterogeneity of the material [25,32].

The work presented in this paper aimed to investigate changes to the mechanical and structural properties of the surface layers of polypropylene samples filled with different talc contents ranging from 5% to 25%. X-ray diffraction was used to determine the crystallographic structure. To characterize the mechanical properties, nano-scratch and ultra-nanoindentation tests were performed at three different depths. The applied methods were used to provide new information needed to understand the role of mineral fillers in the changes to the surface layer of polypropylene polymers.

## 2. Materials and Methods

### 2.1. Sample Preparation

For sample preparation, a highly isotactic polypropylene (PP) was used. A homopolymer with the reference Moplen HP501M was supplied by Basell Orlen Polyolefins, Płock, Poland. The compound was characterized by a density of 0.9 g/cm^3^ (ISO 1183) and the melt flow index MFI_(230 °C, 2.16 kg)_ = 7.8 g/10 min (ISO 1133). As an additive, a talc powder referenced as BP 325 MESH (3MgO 4SiO_2_ H_2_O, TechlandLab LLC, Tarnobrzeg, Poland) was used. The talc density amounted to 2.75–2.80 g/cm^3^ at 20 °C, and the median diameter was 22.4 µm (based on sedimentation analyses).

PP compounds containing 0, 5, 10, 15, 20 and 25 wt.% of talc were prepared by means of the injection-molding process. Before the molding the talc powder and PP pellets were mixed. For preparing the samples, an ARBURG Allrounder 320 injection-molding machine (Arburg GmbH, Lossburg, Germany), equipped with a single-screw plasticizing unit with a screw diameter of 25 mm and a length-to-diameter (L/D) ratio of 20, was used. The clamping force of the mold was 500 kN. The injection pressure was 2000 bar, and the temperature in the plasticizing unit zones had the following values: zone I: 195 °C, zone II: 200 °C, zone III: 210 °C, and zone IV: 220 °C. The specimens for testing were made in a double mold in the shape of tiled beams with cross-sectional dimensions of 4 × 10 mm according to the EN ISO 527-1 standard. As injection molding is one of the most common methods of forming synthetic polymers [33], the device used allowed accurate control of the process parameters and, thus, the samples obtained were repeatable. Therefore, for each combination of PP and talc, one random sample was chosen for testing. The samples were cut off from the middle (Figure 1), and the cross-section planes were trimmed by a microtome (Microm HM355, GMI-Trusted Laboratory Solutions, Ramsey, MN, USA).

### 2.2. Methods

#### 2.2.1. The Scanning Electron Microscope

The obtained samples were sputter coated (Polaron Range S.C. 7620 Sputter Coater, Quorum Technologies Ltd, East Sussex, UK) with a 30 nm layer of gold‒palladium (Au/Pd) and examined using a LEO 1430 VP scanning electron microscope (SEM, Carl Zeiss Microscopy GmbH, Jena, Germany) at an accelerating voltage of 15 kV.

#### 2.2.2. X-ray Diffraction

The crystalline structure was measured by means of the X-ray diffraction (XRD) method. A high-resolution X-ray diffractometer (Empyrean, Malvern Panalytical Ltd, Malvern, UK) with Cu K-alpha radiation (λ = 0.154184 nm) and a Ni filter with a generator voltage of 40 kV and a current of 30 mA was used. The radiation was measured with a proportional detector. The samples were measured in θ–2θ geometry over a range of 10° to 80°. All measurements were carried out at room temperature with a step size of 0.01° and a counting time of 5 s per data point. The source divergence and detector slit were 1/2, and Soller slits were applied. The crystalline phases were identified using the High Score Plus software package (Malvern Panalytical Ltd, Malvern, UK).

#### 2.2.3. Nanoindentation

The nanoindentation tests were carried out at room temperature using the Ultra Nano Hardness Tester (UNHT, Anton Paar GmbH, Graz, Austria) made by CSM Instruments with a diamond Berkovich indenter. The linear loading mode with a maximum depth of 300, 800, and 4000 nm was used. The loading rate was 2 mN/min and a pause of 10 s for 300 and 800 nm was applied. For the depth of 4000 nm the force amounted to 50 mN/min and the pause to 40 s. For the given penetration depth value, ten indentations were made, and the average values of indentation hardness (HIT) and modulus (EIT) were calculated using the Oliver‒Pharr method.

#### 2.2.4. Nano-Scratch Tests

Nano-scratch tests of the investigated materials were carried out using a Nano-Scratch Tester (NST, Anton Paar GmbH, Graz, Austria) made by CSM Instruments equipped with a friction table with a sphero-conical diamond indenter with a radius of 2 μm. The investigations were performed at room temperature along the direction of flow with a multi-pass (3-scan) mode (pre-scan―scratch―post-scan) determining the profile, the depth of the scratch, and the depth after the scratch, respectively. The first scratch test was performed in constant-load mode with the following parameters: constant load 20 mN, horizontal speed 0.2 mm/s, and scratch length 0.8 mm. The second test was performed in progressive-load mode with the following the parameters: progressive loads from 0.2 to 100 mN, loading rate 20 mN/min, horizontal speed 0.3 mm/min, and scratch length 1.5 mm. The applied load in pre-scan and post-scan tests was 0.1 mN. The tests were repeated at least five times for each sample, allowing us to obtain the average dependences of the friction force on the position.

#### 2.2.5. Statistics

Statistical analysis was performed using the Statistica13.1 (TIBCO Software Inc., Palo Alto, CA, USA) and Origin 2016 (OriginLab, Northampton, MA, USA) applications. After the removal of the outliers, the resulting dataset was checked for normality distribution by the Shapiro‒Wilk test and correlations among variables were determined. The two-way analysis of variance (two-way ANOVA) was used to detect significant differences among the tested mechanical parameters (HIT and EIT) depending on the depth at which the measurements were carried out. The following H_0_ hypothesis was tested―there is no difference among the mean values of the determined parameters measured at a particular depth: all μ_ij_ are equal against H_1_: not all μ_ij_ are equal, where µ is the mean value of a determined parameter (HIT, ETI) calculated for the considered group, I (talc content: 0, 5, 10, 15, 20 and 25 wt.%) at a particular depth, j (300, 800 and 4000 nm). After analysis, when there was a basis for rejecting the null hypothesis H_0_, post-hoc tests (Tukey tests) were carried out to identify similar groups. Polynomial contrasts were used to determine the linear effects of increasing talc content and penetration depth on both measured parameters. *P*-values of less than 0.05 were considered as statistically significant.

## 3. Results and Discussion

The SEM images in Figure 2 of the cross section of the samples clearly show that the manufacturing process, in the discussed case of injection molding, facilitated a clear orientation of the filler particles, along the flow direction, relative to the amount of filler introduced into the matrix of the material.

Figure 3a presents the X-ray diffraction profiles measured for pure talc powder―the PP sample and the samples with talc content are presented. For the PP sample, X-ray profiles revealed peaks at 2θ = 14.29°, 17.12°, 18.88° and 21.94°. These peaks corresponded to the (110), (040), (130), and (111) lattice planes of the alpha-crystal PP, respectively. The alpha crystal was arranged in a monoclinic unit cell. In a monoclinic unit, all the axes have different lengths. For the monoclinic form of polypropylene, unit cell parameters were assigned as a = 6.6 Å, b = 20.8 Å, and c = 6.5 Å [34]. On the other hand, talc has a layer structure, in which tetrahedral sheets formed by atoms of silicon and oxygen are covered from above and below by octahedral sheets formed by magnesium hydroxide. This basic structure is characterized by a distance of about 9.5 Å [35]. In the X-ray profile obtained for talc powder, it is apparent that the peak at 2θ = 28.58°was a third-order line of this distance.

Increasing talc content caused the diffraction peaks of the PP crystalline structure to become very weak and completely unnoticeable for samples from 15 wt.%, as shown in Figure 3b. Additionally, the inset in Figure 3b presents dependencies related to the calculated degree of crystallinity. The degree of crystallinity was estimated according to Hulleman et al. [36]. We obtained a lower degree of crystallization than that presented in the literature for buried PP layers owing to the fact that the crystallization process in surface layers was faster due to the direct contact with the injection molds [37]. One can see a significant decrease in the crystallization phase in the surface layer. Our results are in agreement with those obtained by [38,39,40], which confirm the influence of talc on the crystallization process and the crystallinity degree of PP.

In the case of thermoplastic matrices, additives affect the rate of crystallization, the degree of crystallinity, and the nature of the crystalline phase. Due to the fact that the talc constituted a significant percentage of the content, and that the plasticization process took place at temperatures above 200 °C, a strong exfoliation process took place. On one hand, the exfoliation process can lead to the disappearance of the polymeric structure and, on the other hand, the formation of a new phase from the additive. Our XRD measurements for samples with talc suggest the occurrence of this process. For samples with additives, X-ray diffraction profiles did not show peaks from the layered structure of talc. However, they revealed peaks derived from the structure of filler components. The intensity of these peaks rose with the increase of additives in the surface layer of the sample. It was observed that dominant peaks were located at 2θ = 50.29°, 60.81°, and 71.89°.

As the phase analysis showed, the peaks at position 2θ = 50.29° corresponded to the MgOH_2_ (102) lattice plane, as reported by [41,42]. However, the peaks at position 2θ = 60.81° originated from the MgO (331) lattice plane [43]. On the other hand, the peaks at position 2θ = 71.89° originated from the SiO_2_ (020) lattice plane [44,45].

From the ultra nano hardness measurements (Figure 4) changes to the hardness and modulus of the elasticity were obtained (Figure 5a–f and Table 1 and Table 2). Statistical analysis showed that the indentation hardness at a depth of 300 nm related to the talc content did not differ among the groups (Table 1, Figure 5a). For the mentioned depth, a simultaneous slight increase in the elasticity modulus, along with the increasing talc content, can be observed (Table 2, Figure 5b) reaching maximum values for 25 wt.% of talc concentration. The differences in the relation between hardness values and talc content were observed at a depth of 800 nm (Table 1, Figure 5c). The hardness values grew linearly with the talc content from 71 (±3) MPa for pure PP to 83.34 (±5) MPa for PP with 25 wt.%. Pure polypropylene and polypropylene with 25 wt.% talc filler differed significantly in terms of the tested parameters in relation to the other groups. A similar increase can be observed for the EIT values (Table 2, Figure 5d). Statistically significant differences in terms of indentation elasticity can be detected between pure PP, samples with 5 and 10 wt.% talc concentration, and the group of samples with 15–25 wt.% talc concentration, which was uniform in terms of EIT. At a depth of 4000 nm changes both in the indentation hardness and the indentation elasticity were still linear with regard to the talc content (Table 1 and Table 2, Figure 5e,f). In this case, the hardness increase amounted to 18%, while the elasticity modulus changed by about 40% in comparison to pure PP.

Our results indicate that talc addition changed the mechanical properties of PP linearly, which is also proved by the linear contrast *P*-values (*p* < 0.001). The influence of different talc concentrations on the mechanical parameters of polypropylene/talc composites has been widely studied; however, the achieved results show some inconsistencies. The outcomes obtained by L. Lapcik et al. [14] and O. Ammar et al. [46], where the influence of 5 to 40 wt.% talc concentration on the mechanical features of the compounds was studied, showed a similar growing trend in hardness with increasing talc concentration. L. Lapcik et al. [14] registered a gradual increase in microhardness (from 34.94 for pure PP to 38.18 N/mm^2^ for 30 wt.% talc concentration). The mentioned trend was also confirmed by the authors in the corresponding increase of bending strength from the original 21.18 to 39.20 MPa for a 30 wt.% talc concentration. Ammar et al. [46] observed a positive effect with a 30 wt.% talc additive on stiffness and crystallinity, while simultaneously a decrease in impact strength and tenacity was registered. The effects of adding talc-mineral particles of 10 µm diameter with concentrations of 10, 20 and 30 wt.% on the shrinkage and mechanical properties of PP/talc composites were investigated [47]. The research revealed that with the increasing talc content, changes to the measured mechanical parameters were observed. On the other hand, increasing talc concentration caused an increase in the hardness up to some point (6 or 9 wt.%), after which its decrease was noticed [48,49]. The appearance of this type of dependence is explained by the uneven distribution of the filler, as the filler type and its distribution are the most important factors affecting the mechanical properties of PP/talc compounds [48].

Noteworthy are the results calculated for particular talc concentrations and the resultant relation between the measured mechanical parameters and penetration depths. As shown in Table 1 and Table 2 the inverse linear dependence between the hardness and elasticity module and the depth of indentation is recorded. As the depth of indentation increases, a decrease in the values of the tested parameters can be observed. An increasing talc concentration did not influence the general trend observed for HIT and EIT parameters, however for a 25 wt.% talc content the lowest differences in HIT and EIT between particular depths were detected. As mentioned in the introduction section, a wide range of works show that the indentation hardness and elasticity modulus for polymers decrease along with an increasing indentation depth [24,25,26,50]. The observed effect can be caused by various factors, including surface effect, changes in the glass transition temperature, adhesion forces, indentation-tip bluntness, and material inhomogeneity, [24] all being a result of the relatively complex polymer structure and their temperature and pressure-dependence behavior. Taking into consideration the results obtained by other authors [24] it can be concluded that increasing the talc content can influence the homogeneity of the observed composites thereby reducing the extent of the effect of the dependence of the mechanical parameters on the size of the indentation.

Figure 6 shows the average dependence of the friction force on scratch positions for a constant load. For all the samples, a similar increase in the friction-force rate after the start of the scratch test was observed (position 0 to 0.03 mm). Next, the friction force remained stable or grew slightly with the increasing talc content. Although the addition of the talc altered the structure of the PP, enhancing its hardness and elasticity, it did not influence the values of friction force.

Figure 7 presents the dependence of the friction force (a) and penetration depth (b) on the scratch positions for the progressive loads. No differences were found in the value of the friction force in relation to the talc concentration. The penetration depths, in turn, decreased with the talc content for both the constant and progressive loads (Figure 6b and Figure 7b). The observed effect can be associated with the increase in the indentation hardness and the indentation modulus with the growing concentrations of talc in the composites. Research has revealed that talc filling increases the scratch resistance of the PP/talc compound [51], mainly due to the fact that talc particles cause an increase in nuclei density [46,52], which translates into increased hardness. However, in [53] it was shown that talc-filled PP has poor scratch resistance irrespective of the addition of a modifier or a lubricant.

The use of one of many processing methods (extrusion pressing or injection molding) causes the obtained product to have different external and internal characteristics. For a given material, and depending on the type of processing method chosen and the conditions of its implementation, one can obtain the expected top layer, protecting the product against extensively fast surface wear and, thus, protect against damage to the layer continuity.

## 4. Conclusions

To meet demanding engineering and structural specifications, PP is rarely used in its original state and is often transformed into composites by the inclusion of fillers or reinforcements. Our findings have shown that talc, one of the most popular fillers, even in lower concentrations, affected the structural properties of the surface layer of PP‒talc composites. Hardness and elasticity increased along with the talc content, however significant changes were observed for the higher talc filling. Interestingly, for a 25 wt.% talc concentration, the intensity of the size-indentation effect was the lowest, which indicates a more homogenous distribution of talc particles. The increase in talc content did not change the friction on the surface layer of the samples obtained by the injection-molding method, but it influenced the scratch resistance, as measured by the penetration depth. Additionally, XRD measurements revealed that an increase in talc content led to a decrease in the degree of the PP crystallinity of the surface layer and probably caused the exfoliation process in the PP matrix.

## Figures and Tables

**Figure 1 materials-13-00698-f001:**
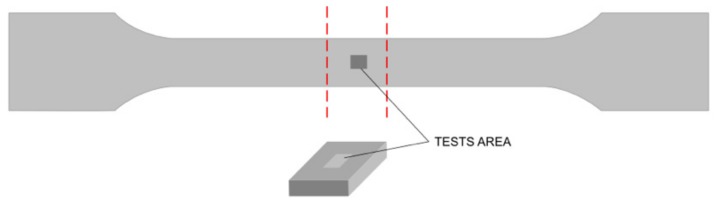
Sample preparation for the tests.

**Figure 2 materials-13-00698-f002:**
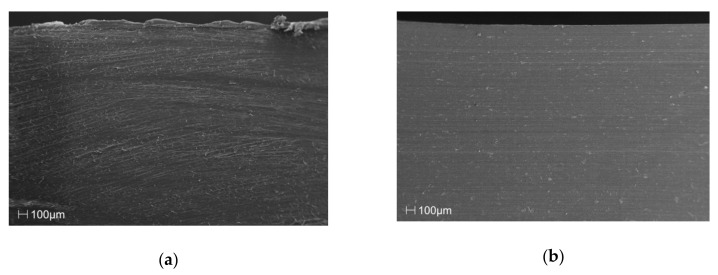
SEM images of the cross-section surface layer for polypropylene (PP) samples with (**a**) 0%, (**b**) 5%, (**c**) 10%, (**d**) 15%, (**e**) 20% and (**f**) 25% talc addition.

**Figure 3 materials-13-00698-f003:**
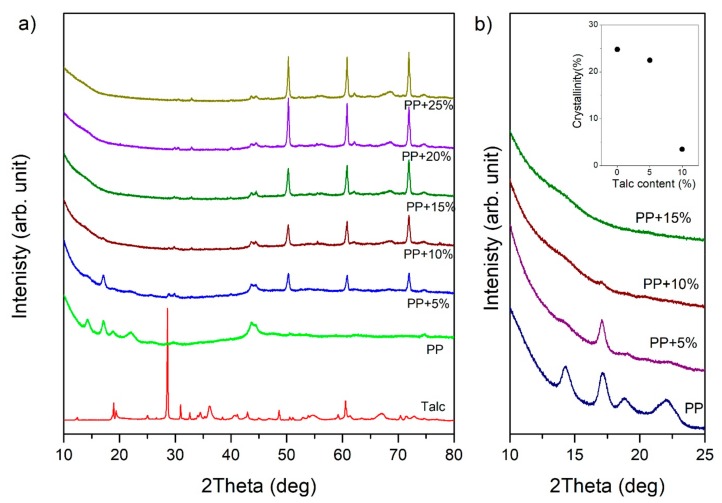
XRD patterns of (**a**) pure talc powder, polypropylene, and samples with different talc content and (**b**) an enlarged range of the diffraction pattern for 2θ from 10° to 25°. The inset presents the relation between the crystallinity degree and the talc content.

**Figure 4 materials-13-00698-f004:**
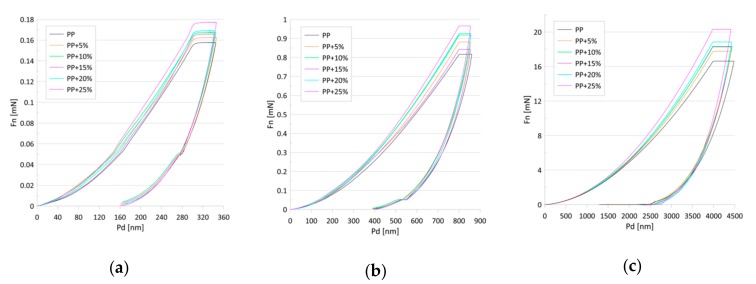
Load (Fn)–penetration depth (Pd) curves for the tested samples for penetration depths of (**a**) 300, (**b**) 800, and (**c**) 4000 nm.

**Figure 5 materials-13-00698-f005:**
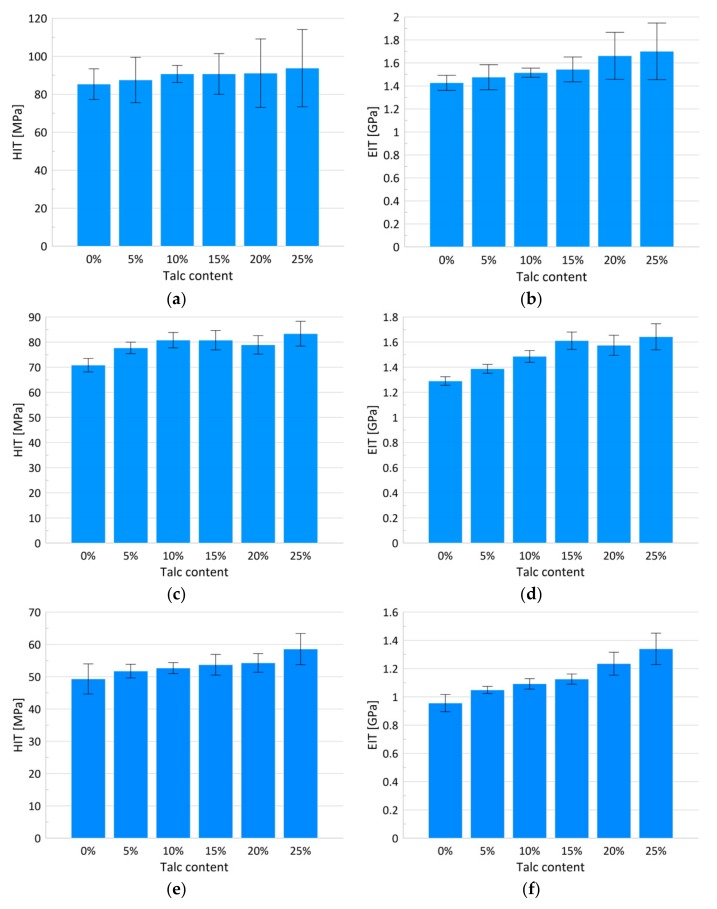
Bar charts of the hardness with corresponding standard deviations for penetration depth of 300, 800 and 4000 nm, respectively (**a**,**c**,**e**); bar charts of the modulus of elasticity with corresponding standard deviations for penetration depth of 300, 800 and 4000 nm, respectively (**b**,**d**,**f**).

**Figure 6 materials-13-00698-f006:**
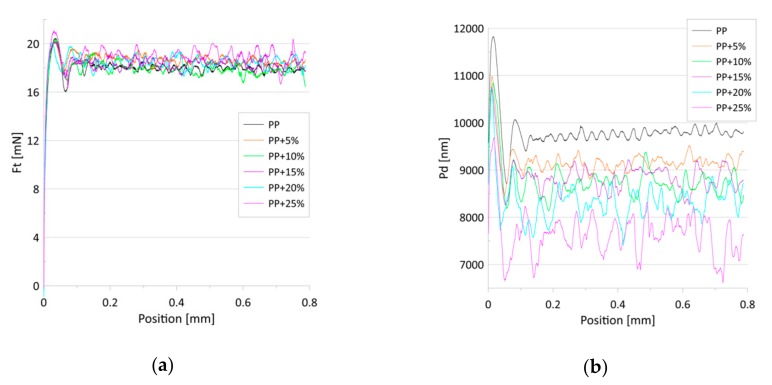
(**a**) The average dependence of the friction force (Ft) on the scratch positions for the tested samples; (**b**) The average dependence of the penetration depth (Pd) on the scratch position for the tested samples. The constant load was 20 mN.

**Figure 7 materials-13-00698-f007:**
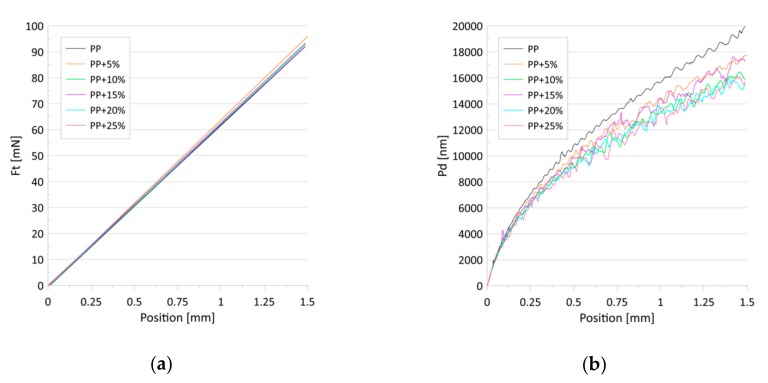
(**a**) The approximated dependence of the friction force (Ft) on the scratch positions for the tested samples; (**b**) The average dependence of the penetration depths (Pd) on the scratch positions for the tested samples. Progressive-load scratch tests were performed for loads from 0.2 to 100 mN.

**Table 1 materials-13-00698-t001:** The mean values of indentation hardness (HIT) measured for different depths with corresponding standard deviations, linear contrast, and post-hoc analysis results.

	HIT (MPa)			*p*-Value (Linear)
Penetration Depth (nm) Talc Content (%)	300	800	4000	
0	85 ^aA^ ± 8	71 ^aB^ ± 3	49 ^aC^ ± 5	<0.001
5	88^aA^ ± 12	78 ^bB^ ± 2	52 ^abC^ ± 2	<0.001
10	91^aA^ ± 4	81 ^bcB^ ± 3	53 ^abC^ ± 2	<0.001
15	91 ^aA^ ± 11	81 ^bcB^ ± 4	52 ^abC^ ± 1	<0.001
20	91 ^aA^ ± 18	79 ^bcB^ ± 4	54 ^bC^ ± 3	<0.001
25	94 ^aA^ ± 20	83 ^cA^ ± 5	59 ^cB^ ± 5	<0.001
*p*-value (linear)	0.012	0.002	0.012	

a, b, c—homogenous group according to talc content for the same penetration depth (at *p* < 0.05). A, B, C—homogenous groups according to penetration depth for the same talc content (at *p* < 0.05)

**Table 2 materials-13-00698-t002:** The mean values of indentation modulus (EIT) measured for different depths with corresponding standard deviations, linear contrasts and post-hoc analysis results.

	EIT (GPa)			*p*-Value (Linear)
Penetration Depth (nm) Talc Content (%)	300	800	4000	
0	1.43 ^aA^ ± 0.07	1.29 ^aB^ ± 0.03	0.96 ^aC^ ± 0.06	<0.001
5	1.48 ^aA^ ± 0.11	1.39 ^bB^ ± 0.04	1.05 ^bC^ ± 0.03	<0.001
10	1.52 ^aA^ ± 0.04	1.49 ^cA^ ± 0.05	1.10 ^bB^ ± 0.01	<0.001
15	1.54 ^abcA^ ± 0.11	1.61 ^dA^ ± 0.02	1.11 ^bB^ ± 0.01	<0.001
20	1.66 ^bcA^ ± 0,20	1.57 ^dA^ ± 0.08	1.24 ^cB^ ± 0.08	<0.001
25	1.70 ^cA^ ± 0.25	1.64 ^dA^ ± 0.10	1.34 ^dC^ ± 0.11	<0.001
*P*-value (linear)	<0.001	<0.001	<0.001	

a, b, c, d—homogenous group according to talc content for the same penetration depth (at *p* < 0.05). A, B, C—homogenous groups according to penetration depth for the same talc content (at *p* < 0.05)

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
