# Peer review of "The Structure and Mechanical Properties of the Surface Layer of Polypropylene Polymers with Talc Additions"

_materials, 2020, doi:10.3390/ma13030698_

Round 1
Reviewer 1 Report
The manuscript “Structure and mechanical properties of polypropylene polymers with talc additions” by MichaÅ‚ Åšwietlicki et al. demonstrates the effect of talc content on the structure, and mechanical properties of polypropylene-based composites. I missed new aspects in this paper. The introduction does not provide a clear background from the relevant studies and the key issues, thus makes it difficult to apprehend the significance of this research. Enough description for the materials and sample preparation is not provided. The discussion only delivers the findings without deep scientific explanation. In many parts the writing should be improved. The novelty of this work is least explained. In summary, I believe this manuscript is not acceptable at its current state without significant modification.
Author Response
Thank you very much for the comments and valuable tips that helped us to improve our work. We are really grateful for the insight you have made into our article. Herein, we offer detailed responses to yours comments.
The introduction does not provide a clear background from the relevant studies and the key issues, thus makes it difficult to apprehend the significance of this research.
The Introduction section (Lines 30-80) was rewritten to provide suitable background on the basis of relevant studies, underlining the novelty of submitted work. First of all, we wanted to pay attention to the effect of dependence of hardness on the size of indentation not studied in the case of PP / talc composites. Remaining issues were changes in the wear resistance and in crystalline structure of PP/talc materials with increasing talc content.
Enough description for the materials and sample preparation is not provided.
Extended description of the materials used and the process of sample preparation was added (Lines 83-88 and 97-101)
The discussion only delivers the findings without deep scientific explanation.
Thank you for the comment. In the revised version of the manuscript, we have made an attempted to explain each of the observed phenomena on the basis and in the relation to the previous findings and research results (Results and discussion section).
In many parts the writing should be improved.
The necessary language corrections have been made.
The novelty of this work is least explained.
As mentioned in the answer to the first question, the novelty relied on the looking at PP/talc composites through the prism of mechanical properties of the surface as an indicator allowing to understand the role of mineral fillers in the changes of the surface layer of polypropylene polymers. We wanted to pay a special attention to the indentation size effect, which, to the best of our knowledge, has not been studied yet for the polypropylene and talc composites.
Reviewer 2 Report
I think that the subject has been investigated quite thoroughly already. Here are my detailed comments:
L21: What does this indicate regarding structure and/or mechanical properties? Add some explanation.
L67: How was degradation process discussed in this paper? Add it to the text in clear form.
L76: How many specimen were analyzed per material for each test?
L106: It has been noted that crystals & filler particles may align differently regarding the flow in injection molding - were the measurements made along the direction of flow?
L128: "technological process" or just manufacturing process?
L139: You could have measured the changes in crystallinity caused by talc addition using XRD at least at the lower levels.
L142: Where are "chain orientation, crystal type and degree of crystallinity presented"? Please add them with some discussion.
L144: Include XRD spectra of pure talc also.
L160: "PP+25%" - use consistent labeling along the paper.
L175: I think you already stated that your findings are in agreement with Lapic et al in L173.
L177: What does the "Mentioned trend" exactly refer to?
L189: Mentions of impact strength and tensile modulus makes me want to see your results..
L186: Make the bottom halves of standard deviations also visible.
L192: Where is tensile strength presented?
L201: Could changes in crystalline structure/crystallinity have an effect? Caused e.g. by gradient in cooling rate. (I do not know, just speculating)
L209: "did not increase the resistance" but in L218 "penetration depth decreased". Why is penetration depth not considered an indicator of scratch resistance?
L229: Any XRD findings worth mentioning in Conclusions? You do mention something in the Abstract.
Author Response
Thank you very much for the comments and the insight you have made into our article. Below we offer detailed responses to yours comments.
L21: What does this indicate regarding structure and/or mechanical properties? Add some explanation.
The Abstract was completely rewritten. Corresponding sentence may be found in lines 25-27.
L67: How was degradation process discussed in this paper? Add it to the text in clear form.
The degradation process was not studied in the presented paper. The sentence referred to the wear of the elements made of polymers, especially polypropylene. For the greater clarity the sentence has been rewritten (Lines 79-80)
L76: How many specimen were analyzed per material for each test?
Due to the repeatability of the sample production process for each combination of PP and talc one random sample was chosen for the testing.
L106: It has been noted that crystals & filler particles may align differently regarding the flow in injection molding - were the measurements made along the direction of flow?
Of course we agree with this statement, which is why the measurements were made along the direction of flow (Line 131)
L128: "technological process" or just manufacturing process?
The expression has been corrected (Line 156)
L139: You could have measured the changes in crystallinity caused by talc addition using XRD at least at the lower levels.
The Results and discussion section have been completely rewritten in the range of XRD measurements.
L142: Where are "chain orientation, crystal type and degree of crystallinity presented"? Please add them with some discussion.
The whole paragraph in the Results and discussion section was rewritten and explanation concerning XRD results may be found in Lines 161-195.
L144: Include XRD spectra of pure talc also.
The XRD spectra of pure talc has been added (Figure 3a).
L160: "PP+25%" - use consistent labeling along the paper.
The expression has been uniformed along the paper
L175: I think you already stated that your findings are in agreement with Lapic et al in L173.
The whole paragraph has been rewritten and repeating expressions has been removed.
L177: What does the "Mentioned trend" exactly refer to?
The “mentioned trend” (Line 225) refers to the “growing trend” from the previous sentence.
L189: Mentions of impact strength and tensile modulus makes me want to see your results.
Thank you for the comment. In the Results and discussion section we have tried to concentrate on the results from comparable tests.
L186: Make the bottom halves of standard deviations also visible.
The standard error bars were shown in both directions.
L192: Where is tensile strength presented?
As the tensile strength was not measured the results cannot be presented in that work. Mentioned fragment has been rewritten.
L201: Could changes in crystalline structure/crystallinity have an effect? Caused e.g. by gradient in cooling rate. (I do not know, just speculating)
The whole paragraph in the Results and discussion section was rewritten and explanation concerning XRD results may be found in Lines 161-195.
L209: "did not increase the resistance" but in L218 "penetration depth decreased". Why is penetration depth not considered an indicator of scratch resistance?
Thank you for the comment. As suggested, the penetration depth was considered as an indicator of scratch resistance and the paragraph was reedited (Lines 265).
L229: Any XRD findings worth mentioning in Conclusions? You do mention something in the Abstract.
Conclusion has been rewritten and XRD results were provided (Lines 290-300)
Reviewer 3 Report
General comments:
The authors present an interesting study on the structural and mechanical changes of PP via addition of talc. The experimental set-up is comprehensible and the results are presented in an appropriate way. However, the manuscript has two major flaws, which should be addressed before publication:
The language style of the manuscript is insufficient. The content becomes clear after rereading sentences several times, but the poor grammar prevents fluid reading, and thus, distracts from the scientific content. The reviewer suggests a professional proofread. Another concern is the apparent lack of novelty. The authors describe their findings and relate it to content already present in the literature. This is comprehensible and should be part of every scientific paper. However, the authors do not state, which of their results are novel and what their manuscript might be able to add to the state of the current knowledge. The reviewer suggests to add paragraphs to highlight the novelty in the results part as well as in the conclusion.
Other questions/comments:
Line 21: The meaning of this sentence is not clear. Line 50: “These tests….” Which tests are the authors referring to? Line 160/176/178: Errors should be presented in the text to allow a comparison of the discussed values. Table 1: Considering the error values, it is not necessary to present any positions after the decimal point for the HIT values in this table.
Minor spell check:
Line 14: Wrong formula: 3 MgO * 4 SiO2 * H2O, “2” is missing in SiO2 and both “2” should be subscripted. Line 56: This sentence is missing a verb. Line 73/162/189: Please use non-breaking space to prevent the separation of numbers and the respective units over two lines. Line 171: “talk” instead of “talc”. Line 202: Missing period after citations. Line 215/223: A period should be used instead of a semicolon at the end of the caption. Line 286: Author names are completely in capital letters, unlike the rest of the references.Author Response
Thank you very much for the comments that allow us to improve our submission.
The language style of the manuscript is insufficient. The content becomes clear after rereading sentences several times, but the poor grammar prevents fluid reading, and thus, distracts from the scientific content. The reviewer suggests a professional proofread.
Language corrections have been made.
Another concern is the apparent lack of novelty. The authors describe their findings and relate it to content already present in the literature. This is comprehensible and should be part of every scientific paper. However, the authors do not state, which of their results are novel and what their manuscript might be able to add to the state of the current knowledge. The reviewer suggests to add paragraphs to highlight the novelty in the results part as well as in the conclusion.
Thank you for the comment. In the revised version of the manuscript we have tried to underline the novelty of our findings, especially concerning the indentation size effect, which, to the best of our knowledge, has not been studied yet for the polypropylene and talc composites. We want to look at PP/talc composites through the prism of mechanical properties of the surface as an indicator allowing to understand the role of mineral fillers in the changes of the surface layer of polypropylene polymers.
Other questions/comments:
Line 21: The meaning of this sentence is not clear.
The Abstract was completely rewritten. Corresponding sentence may be found in lines 25-27.
Line 50: “These tests….” Which tests are the authors referring to?
After modifications of an Introduction section the sentence was removed.
Line 160/176/178: Errors should be presented in the text to allow a comparison of the discussed values.
Errors mentioned to be missing for the line 160 were added (Line 206)
The values provided in a lines 176 and 178 came from the work entitled “Effect of the Talc Filler Content on the Mechanical Properties of Polypropylene Composites”. The fact of citation was underlined in the remaining sentence (Line 226). We are not able to provide the error values as no information about them is added in the mentioned article.
Table 1: Considering the error values, it is not necessary to present any positions after the decimal point for the HIT values in this table.
The values in the table 1 have been changed according to the Reviewer’s suggestion.
Minor spell check:
Line 14: Wrong formula: 3 MgO * 4 SiO2 * H2O, “2” is missing in SiO2 and both “2” should be subscripted.
The formula has been corrected (Line 14)
Line 56: This sentence is missing a verb.
The sentence has been corrected
Line 73/162/189: Please use non-breaking space to prevent the separation of numbers and the respective units over two lines.
Non-breaking spaces have been applied throughout the work.
Line 171: “talk” instead of “talc”.
The error has been corrected. (Line 220)
Line 202: Missing period after citations.
The period has been added. (Line 255)
Line 215/223: A period should be used instead of a semicolon at the end of the caption.
The period instead of semicolon was used (Line 270 and Line 283).
Line 286: Author names are completely in capital letters, unlike the rest of the references.
The record has been corrected.
Round 2
Reviewer 1 Report
The revised version of the manuscript “Structure and mechanical properties of polypropylene polymers with talc additions” by MichaÅ‚ Åšwietlicki et al. reflects the comments made by the reviewers. But I still have the following comments before the acceptance of the manuscript for publication.
The first paragraph of the Introduction, where a direct statement is provided in favour of polypropylene (PP) as a potential substitute for polyamide in many specialty applications is quite misleading. Please check the following paper (Journal of Colloid and Interface Science 403 (2013) 29–42). Also, reference 2 is not appropriate in this context. Lines 38-39, Change “PP is isotactic” to “Isotactic PP is”. In Figure 3b, why the crystallinity of neat PP is much lower than the usual crystallinity of isotactic PP (40-50%)? Please comment. Lines 183-184, there is a mistake in counting “a peaks”.Author Response
Thank you very much for the comments that allow us to improve our submission.
The first paragraph of the Introduction, where a direct statement is provided in favour of polypropylene (PP) as a potential substitute for polyamide in many speciality applications is quite misleading. Please check the following paper (Journal of Colloid and Interface Science 403 (2013) 29–42). Also, reference 2 is not appropriate in this context.
Thank you for the comment. In the first paragraph of the Introduction section we wanted to underline the possibility of replacing polyamide with polypropylene in a humid environment.
Lines 38-39, Change “PP is isotactic” to “Isotactic PP is”.
The sentence has been corrected.
In Figure 3b, why the crystallinity of neat PP is much lower than the usual crystallinity of isotactic PP (40-50%)? Please comment.
Corresponding sentence may be found in lines 177-179
Lines 183-184, there is a mistake in counting “a peaks”.
The mistake has been corrected. (Lines 189-190)
Reviewer 2 Report
You have improved the paper notably, only thing I really noticed was that L 209 seem to contain a typo ".. terms of ETI."
Author Response
Thank you for the comment.
You have improved the paper notably, only thing I really noticed was that L 209 seem to contain a typo ".. terms of ETI."
The mistake has been corrected. Line 214
Reviewer 3 Report
The revised version of the manuscript was significantly improved. The problems regarding the language style and the apparent lack of novelty were addressed accordingly. The reviewer suggests to accept the manuscript for publication once the minor errors (see below) were addressed.
Minor spell check:
Line 31/54: “machine parts” instead of “machines parts”, remove the first “s”
Line 36/37: “(PP)” should be moved to line 36, were “polypropylene” was mentioned the first time in the text.
Line 86: Missing space in the formula between SiO2 and H2O
Line 93/122/223/225: Please use non-breaking space to prevent the separation of numbers and the respective units over two lines.
Line 122: “pause of a 10 s for 300 nm and”, remove the “a”
Line 184: “they revealed a peaks derived”, remove the “a”
Line 225: “the Authors“, „authors“ should not start with a capital letter
Line 255: “the Introduction section”, „introduction“ should not start with a capital letter
Line 256: “works shows that the”, “show” without the “s”
Author Response
Line 31/54: “machine parts” instead of “machines parts”, remove the first “s”
The mistake has been corrected. Lines 32 and 56
Line 36/37: “(PP)” should be moved to line 36, were “polypropylene” was mentioned the first time in the text.
“(PP)” has been moved to line 37.
Line 86: Missing space in the formula between SiO2 and H2O
Missing space has been added.
Line 93/122/223/225: Please use non-breaking space to prevent the separation of numbers and the respective units over two lines.
Non-breaking spaces have been added.
Thank you very much for the comments that allow us to improve our submission.
Line 122: “pause of a 10 s for 300 nm and”, remove the “a”
The sentence has been corrected.
Line 184: “they revealed a peaks derived”, remove the “a”
The sentence has been corrected.
Line 225: “the Authors“, „authors“ should not start with a capital letter
The sentence has been corrected. Line 234
Line 255: “the Introduction section”, „introduction“ should not start with a capital letter
The sentence has been corrected. Line 259
Line 256: “works shows that the”, “show” without the “s”
The sentence has been corrected. Line 260